# An Outlier Cleaning Based Adaptive Recognition Method for Degradation Stage of Bearings

**DOI:** 10.3390/s22176480

**Published:** 2022-08-28

**Authors:** Jingsong Xie, Yujie Xie, Tiantian Wang, Yougang Xiao

**Affiliations:** 1School of Traffic and Transportation Engineering, Central South University, Changsha 410075, China; 2School of Mechanical and Vehicle Engineering, Hunan University, Changsha 410082, China

**Keywords:** degradation stage recognition, outlier cleaning, accurate impulse locating, RUL prediction

## Abstract

Accurate identification of the degradation stage is key to the prediction of the remaining useful life (RUL) of bearings. The 3σ method is commonly used to identify the degradation point. However, the recognition accuracy is seriously disturbed by the random outliers in the normal stage. Therefore, this paper proposes an adaptive recognition method for the degradation stage based on outlier cleaning. Firstly, an improved multi-scale kernel regression outlier detection method is adopted to roughly search the abnormal signal segments. Then, a method for the accurate locating of the start and end points of abnormal impulses is established. After that, indexes are constructed for screening abnormal segments and an iterative strategy is proposed to achieve an accurate and efficient removal of abnormal impulses. After outlier cleaning, the 3σ approach is used to set the degradation warning threshold adaptively to realize the degradation stage recognition of the bearings. The PHM 2012 rotating machinery dataset is used to verify the effectiveness of the proposed method. Experimental results show that the proposed method can accurately locate and remove the outliers adaptively. After the cleaning of the outliers, the identification of the degradation stage is no longer disturbed by the selection of the reference signal of the normal stage and the robustness and the accuracy of the degradation stage identification have been improved significantly.

## 1. Introduction

Prognostics and health management (PHM) is an important technology in modern industries that can enhance the reliability of mechanical equipment, reduce unnecessary maintenance costs and avoid catastrophic accidents [1,2,3]. The performance of bearings (one of the key components of rotating machinery) directly affects the safe and reliable operation of mechanical equipment. In order to maintain running time and to prevent abnormal failure, it is necessary to monitor the performance degradation of bearings in the operation state.

Generally, the degradation process of bearings can be divided into two stages, namely the normal stage and the irreversible degradation stage [4]. Ideally, in the normal operation stage, the bearing is in a healthy state and the monitoring data fluctuate randomly in a small range with no significant difference. In the irreversible degradation stage, the bearing will present early failure and then begin to degrade, and the degradation degree becomes more and more serious as the running time increases. In order to estimate the degradation accurately and to improve the prediction accuracy of the RUL, different methods were proposed. In some research [5,6], scholars artificially and subjectively determined the degradation stage of bearings. Li et al. [7] determined that when the health index of bearings exceeded the 3σ interval, this was the dividing point between the normal and the degradation stage. Kong et al. [8] proposed an adaptive time selection method for the first prediction based on the 3σ approach. Singh et al. [9] applied Chebyshev’s inequality to the identification of the locations of the bearing health state change points.

In the above 3σ approach, a piece of healthy signal data is selected as the standard to build the 3σ threshold for the degradation warning. However, due to inappropriate selections of the health stage, it is easy to make an identification error of the degradation stage. Therefore, Pan et al. [10] determined the degradation threshold according to the continuous gradient value of health indicators with running time. Gao et al. [11] proposed a change point degradation model with random jump and considered a random change point that was influenced by the accumulative effect of a shock process. In their work, human involvement and the application of professional knowledge are required. Different working conditions will bring various performance degradation trends of bearings. Therefore, it is not universal to choose fixed parameters under specific working conditions. In addition, some scholars introduced the neural network into the recognition of the bearing’s degradation stage. Li et al. [12] generated an adjournment network to learn the data distribution of bearing health stages to determine the change point of the degradation state. Baptista et al. [13] used a recursive neural network to find the location of the degradation point. Wu et al. [14] classified a system health stage based on the trained model and real-time sensory data. However, in actual work, vibration signals in the normal operation stage of bearings will be covered by noise and random abnormal shocks [15] and the constructed degradation health indicators show serious shocks and poor stability and monotonicity, thus affecting the accurate recognition of the degradation stage.

In order to reduce the influence of random abnormal impulses in the normal operation stage, scholars used empirical mode decomposition (EDM), wavelet threshold denoising, Fourier transform (FFT) and other signal processing methods to reduce the noise of bearing vibration signals [16,17,18]. Although such signal processing methods may work well in some cases, they will weaken the original signal to some extent or remove some key signals altogether, which may lead to low prediction accuracy. In addition, Li et al. [19] used the exponential weighted moving average (EWMA) method to smooth the health indicators. If the oscillation and deviation of the current point were large, the EWMA method could not obtain a good smoothing effect. Chen et al. [20] proposed a fault impulse method named the Intelligent Impulse Finder to accurately identify and locate the fault impulses; however, there are cases of missed or false detections. M. Gupta et al. [21] used the outlier detection method to look for local abnormal areas, but since the detected abnormal areas were formed by some continuous outlier points, which sometimes did not exist, the outlier detection method could only eliminate the oscillation phenomenon of health indicators locally according to the outlier area.

To solve the above problems, this paper proposes a degradation stage recognition method based on outlier cleaning. In the global perspective, an improved outlier detection method based on the local outlier factor is used to detect abnormal signal segments adaptively. For the local abnormal signal segment, according to the kernel density estimation and the analysis of the gradient value of the abnormal impulse envelope gradient value to achieve an accurate impulse location. Then, a screening index of abnormal signal segments and an iterative removal strategy are constructed to removing abnormal impulses, thereby reducing the time and obtaining a better effect for data cleaning. Finally, a health index representing the degradation characteristics of the mechanical equipment is constructed for the vibration signals after abnormal processing. The degradation threshold set by the 3σ method is used to identify the degradation stage of the bearings.

The main contributions are as follows:(1)An outlier detection method is constructed by combining the global search of abnormal signal segments and the accurate locating of local abnormal impulses. Adopting the constructed method, the normal signals misjudged as outliers in abnormal signal segments can be greatly reduced and the normal data can be fully retained.(2)A screening criteria and an iteration removal strategy are proposed, achieving an adaptive detection of segments that contain abnormal impulses and a quick removal of abnormal segments.(3)After outlier cleaning, the 3σ approach is used to set the degradation warning threshold adaptively to realize the degradation stage recognition of bearings. The outlier cleaning avoids the interference of the selection of reference signals and improves the accuracy and stability of the degradation stage recognition.

This paper is organized as follows: Section 2 describes the algorithm principle and the method flow of the proposed method. In Section 3, the proposed method is verified by the degradation data of rolling bearings. Some conclusions are made in Section 4.

## 2. The Proposed Method

The proposed method is shown in Figure 1, including four key steps: (1) global detection of abnormal signal segments; (2) accurate locating of local abnormal impulses; (3) outlier removal method; (4) degradation point adaptive detection.

### 2.1. Detection Method for Abnormal Signal Segments

The kernel regression based abnormal detection method is adopted to search for abnormal signal segments globally that are contained in the degraded full life data. Firstly, the time–domain statistical features are extracted from the raw vibration signals. Then, the local outlier factor of these features based on kernel regression are calculated to assess the abnormity degree of each data segment. Finally, the abnormal data segments can be detected according to the local outlier factor and the anomaly determination threshold.

#### 2.1.1. Data Preprocessing

The raw vibration signal data are divided into segments by the sliding window, and the time–domain features are extracted from each segment. In order to obtain great detection performance, appropriate features should be selected to constitute a feature set for calculating the local outlier factor. The features adopted include mean, maximum, minimum, peak-to-peak, wave peak factor, variance, root mean square, mean amplitude, root amplitude, waveform factor, impulse factor, margin factor, kurtosis, peak and skewness [22].

#### 2.1.2. Local Outlier Factor

The local outlier factor detects outliers by comparing the density of each sample in the dataset with the density of sample points in the field. The local outlier factor of a point *p* in the dataset is calculated as follows:(1)$$LOF_{k}=\frac{\sum_{o\in N_{k}\left(p\right)}\frac{lrd_{k}\left(o\right)}{lrd_{k}\left(p\right)}}{\left|N_{k}\left(p\right)\right|}=\frac{\sum_{o\in N_{k}\left(p\right)}lrd_{k}\left(o\right)}{\left|N_{k}\left(p\right)\right|}/lrd_{k}\left(p\right)$$
where *k* is the *k*th neighborhood of *p*, and *o* is a point in the *k*th neighborhood of *p*. *lrd*_k_ (·) is the locally accessible density; it is obtained according to:(2)$$lrd_{k}\left(p\right)=1/\left(\frac{\sum_{o\in N_{k}\left(p\right)}reach-dist_{k}\left(p,o\right)}{\left|N_{k}\left(p\right)\right|}\right)$$
where *N_k_*(*p*) is the *k*th distance neighborhood of *p*: *N_k_*(*p*) = {*q*∈*D*|*d*(*p*, *q*) ≤ *kd*(*p*)}; reach-*dist_k_*(*p*, *o*) is the reachability distance of the point *p* to *o*: reach-*dist_k_*(*p*, *o*) = max{*kd*(*o*), *d*(*p*, *o*)}; *kd*(*p*) is the *k*th distance between *p* and a point *o*
∈
*D*, which satisfies the following two criteria: (1) there are at least *k* objects *q*, such that *d*(*p*, *q*) ≤ *d*(*p*, *o*); (2) there are at most *k* − 1 objects *q*, such that *d*(*p*, *q*) < *d*(*p*, *o*). *d*(*p*, *q*) is the distance between point *p* and *q* and defined as:(3)$$d\left(p,o\right)=\sqrt{\sum_{i=1}^{\dim}{\left(f_{i}\left(p\right)-f_{i}\left(o\right)\right)}^{2}}$$
where *f*_i_(*p*) and *f*_i_(*o*) are the time–domain features of points *p* and *o,* respectively, and dim is the dimension of features. Some basic concepts in this expression can be referred to in the paper [23].

#### 2.1.3. Local Outlier Factor Based on Multi-Scale Kernel Regression

The detection of outliers by the local outlier factor method is based on local density estimation, which indicates that a good detection performance requires an accurate density estimation. The outlier method roughly estimates the local density by calculating the reciprocal of the average reachable distance between a given object and its neighborhood points; hence, the method may fail to detect true outliers. Moreover, the local outlier factor is calculated in the *k*-distance neighborhood, so the local outlier factor is very sensitive to parameter *k*. If the outliers are randomly distributed in several clusters with different densities, the local outlier factor method may obtain poor outlier detection results because the parameter *k* cannot meet the detection requirements of all outliers. In order to overcome the above shortcomings, Nadaraya–Watson kernel regression can be used to calculate regression estimators in the neighborhood of *k* distance to construct a more robust local outlier factor [23] in order to construct the final index for anomaly evaluation. Based on the Nadaraya–Watson kernel regression, the local outlier factor expression of *p* is calculated as follows:(4)$$LOF_{s}\left(p\right)=\frac{\sum_{o\in N_{k}\left(p\right)}\frac{1}{{\left(kd\left(o\right)\right)}^{\gamma }}K\left(\frac{d\left(p,o\right)}{kd\left(o\right)}\right)LOF_{k}\left(o\right)}{\sum_{o\in N_{k}\left(p\right)}\frac{1}{{\left(kd\left(o\right)\right)}^{\gamma }}K\left(\frac{d\left(p,o\right)}{kd\left(o\right)}\right)}$$
where *s* is the step of iteration; *γ* is the sensitivity, which defaults to 2; *d*(*p*, *o*) is the distance between point *p* and *o*; *kd*(·) is the *k*th distance; *LOF*_k_(·) is a local outlier factor; *K*(·) is the kernel function, and the kernel function of parameter *x* is shown as follows:(5)$$K\left(x\right)=\{\begin{array}{cc} \beta , & if\left\|x\right\|\leq 1\\ \beta \cdot \exp\left(-{\left(\left\|x\right\|-1\right)}^{2}/2\right), & otherwise \end{array}$$
where *β* is a constant and is set to 1.

#### 2.1.4. Outliers Determination

The abnormal index is constructed based on the multi-scale kernel regression function representing the deviation degree from the normal sample. During the normal operation stage of bearings, most of the signal data are in a normal state and belong to the same cluster. The index value of the abnormal data is much higher than that of the normal data. In order to detect outliers more accurately, when the outlier index is greater than the set anomaly threshold *T*, it is judged as an outlier.
(6)T=μ+3σ
where *μ* is the average value and *σ* is the standard deviation.

### 2.2. Accurate Locating Method for Abnormal Impulses

The abnormal signal segments obtained from the global perspective contain a large number of normal signal data and a small number of abnormal impulses. In order to retain more effective raw vibration signal data, the impulse contained in the abnormal segments will be located accurately by the determining of the start and end points.

#### 2.2.1. Start Point Identification Methods for Impulses

In the abnormal signal segment, the distribution of the normal signal and the abnormal impulse is unknown. In order to estimate the distribution of the population sample data in the abnormal signal segment, the kernel density estimation is used to estimate the probability distribution of the normal signal and the abnormal impulse. In the degradation process of the bearings, the local extreme value of the outlier is stronger than that of the signal in the normal stage, so the local extreme value can well reflect the difference between the normal signal and the abnormal impulse.

First, the local extreme values (*x*_1_, *x*_2_, …, *x_n_*) for each sliding window of the abnormal signal segments are extracted. Then, the kernel density estimate of the extreme value sequence is calculated. The formula of the kernel density estimate is as follows: (7)$${\hat{f}}_{h}\left(x\right)=\frac{1}{nh}\sum_{i=1}^{n}K\left(\frac{x-x_{i}}{h}\right)$$
where *h* is the bandwidth and *K*(·) is the kernel function and
(8)$$K\left(X\right)=\frac{1}{\sqrt{2\pi }}\exp\left(-\frac{1}{2}X^{2}\right)$$
(9)$$h=c\cdot n^{-1/5}$$
where *c* = 1.05 **σ* and *σ* is the standard deviation [24].

Because the abnormal signal segment contains a normal signal and an abnormal impulse, the probability distribution calculated by the kernel density estimation shows a bimodal distribution [25]. Therefore, the minimum at the trough between these two peaks is regarded as the threshold for separating the normal signal and the abnormal impulse. When the local extremum exceeds the threshold for the first time, the corresponding position is judged as the start point of the abnormal impulse.

#### 2.2.2. End Point Identification Methods for Impulses

The amplitude of an impulse rises sharply after the start point and then is attenuated in an exponential form approximately. Therefore, the envelope of the abnormal impulse can be fitted by the exponential function. In order to determine the end point of the abnormal impulse, Hilbert transform is adopted to obtain the envelope of the impulse and then the exponential function is used to fit the envelope.

The end point of the abnormal impulse is considered as the point where the gradient of the envelope is less than the critical value. Therefore, the end points of the abnormal impulse are determined by analyzing the gradient of the fitting function.

With the step size of 1, the gradient value of the signal segments with n points of the fitting function is calculated successively:(10)$$g_{i}=\frac{y_{i}\left(n\right)-y_{i}\left(1\right)}{n-1}$$
where *y_i_* is the *i*-th signal segment of the fitting function and *g_i_* is the gradien of the *i*-th signal segment. When the gradient *g_i_* reaches the set threshold, the corresponding position is considered as the end point of the abnormal impulse.

### 2.3. Removing Method for Outliers

After the detection of the abnormal signal segments and the accurate locating of impulses, how to remove the outliers efficiently and accurately is the key problem to be solved.

The local outlier factor detection method based on kernel regression can only detect the abnormal signal segments that have a higher anomaly degree compared with normal data relatively, and it cannot identify all abnormal segments in one detection process. In addition, the detection effect of the outliers depends on the parameters of the sliding window and the *k* value of the *k*-distance neighborhood. If the parameter steps are too small in the search process, the calculation efficiency will be reduced and the time of outlier cleaning will be greatly increased.

Therefore, the iterative removal strategy is proposed to overcome the above difficulties. As shown in Figure 2, the window length w and the *k* value are constantly updated in each iteration to obtain the optimum parameter. In this strategy, the detection accuracy and the detection efficiency of the abnormal signal segments are improved.

The abnormal segments identified by the kernel-regression-based method are not only the abnormal segments of impulse types that have significant interference with the identification of degradation stage but also other types of abnormal signal segments with almost no interference in the identification of the degradation stage. Therefore, a screening criteria is constructed to realize the selection of abnormal signals, so that only the abnormal signal segments containing significant impulses are selected for further locating, thus avoiding the time-consuming process of locating other abnormal signal segments with insignificant impulses.

The screening criteria for impulse-type abnormal signal segments are as follows, that is, the maximum vibration signals in an abnormal segment should be greater than that of the vibration signals in the optimal sliding window length at the two adjacent ends:(11)$$\begin{array}{l} \max\left(\{x_{1},x_{2},\cdots ,x_{n}\}w_{i}\right)\geq \max\left(\{x_{1},x_{2},\cdots ,x_{n}\}w_{i-1}\right)\\ \max\left(\{x_{1},x_{2},\cdots ,x_{n}\}w_{i}\right)\geq \max\left(\{x_{1},x_{2},\cdots ,x_{n}\}w_{i+1}\right) \end{array}$$
where $\left\{x_{1},x_{2},\cdots ,x_{n}\right\}$ is the vibration signal data of the bearings; *n* is the sliding window length; $\{\cdot \}w_{i}$ is the *i*-th sliding window; max(⋅) is the maximum.

Finally, using the abnormal signal segment detection method and the screening criteria for impulse-type abnormal signal segments, combined with the accurate locating method for impulses and the iterative removal strategy, the impulse-type outliers with significant interference in the identification of the degradation stage can be cleared efficiently and accurately.

### 2.4. Degradation Point Detection

The 3*σ* approach has been widely used in the degradation stage recognition of bearings [26] and it can set the degradation warning threshold adaptively according to the degradation characteristics of the bearing itself. The method is based on the theory of Chebyshev inequality [27]:(12)$$P\left(\left|X-\mu \right|\geq K\sigma \right)\leq \frac{1}{K^{2}}$$
where *P* is the occurrence probability of event; *X* is a random variable; *K* is any positive real number; *μ* is the mean; σ is the standard deviation.

According to Formula (12), the probability of sample points in the random variable *X* falling outside *Kσ* is not greater than 1/*K*^2^.

Therefore, it can be inferred from Formula (12) that when *K* is set to 3, the probability of any sample falling outside the interval of 3*σ* (i.e., [μ−3σ,μ+3σ]) is less than or equal to 1/9. At the same time, in order to further limit the interference caused by random anomalies, an improved 3*σ* strategy is adopted [8], that is, ti is the degradation point only when the following conditions is satisfied:(13)$$\left|X\left(t_{i}\right)-\mu \right|\geq 3\sigma ,i=l,l+1,\cdots ,l+v-1$$
where *v* is usually taken as 5, *μ* is the mean and *σ* is the standard deviation of the reference signals in the normal stage.

In the detection of the degradation point of bearings, the health indicators should be constructed first. Based on the improved 3*σ* approach, when the health indicators exceed the 3*σ* interval for five times in a row, the corresponding point can be detected as the degradation point to divide the normal and the degradation stage.

The algorithm flow of degradation stage recognition based on outlier cleaning is shown in Figure 2. In Stage 1, the parameters that affect the detection results of the abnormal signal segments include *k* in the local outlier factor and the sliding window length *w*. *k*1 denotes the initial iteration value of parameter *k*, and *k_max_* denotes the maximum of parameter *k*. The initial window length *w*1 is determined as 500 and the step length is 20% of the window length. *w*2 denotes the maximum of the sliding window and it depends on the length of the whole data l: *w*2 = *l*/*k*1. These parameters are constantly updated to search for the best abnormal signal segments from a global perspective. In Stage 2, if the signal segments meet the screening conditions of the impulse-type abnormal signal segments, the start and end points of the abnormal impulse can be further determined.

## 3. Experiments and Analysis

This section may be divided by subheadings. It should provide a concise and precise description of the experimental results, their interpretation, as well as the experimental conclusions that can be drawn.

In this section, a PHM 2012 dataset is used to verify the proposed degradation stage recognition method. The detection effects of outliers and the degradation stage are compared with that of the current methods.

### 3.1. PHM 2012 DataSet

The PHM 2012 challenge dataset used in this study was collected on a PRONOSTIA platform that accelerated the degradation of bearings, as shown in Figure 3. The motor, bearing, gearbox and two pulley assemblies were used to change the speed of the rolling bearing. The acceleration sensor was fixed on the outer ring of the bearing and collected vibration signals in the horizontal and vertical directions, respectively. Similarly, only horizontal vibration data are used in this study. This dataset contains 17 bearing failure data from run to failure under different working conditions [28]. The sampling frequency is 25.6 kHz and 2560 data points are collected every 10 s. For safety, the test was stopped when the amplitude of the vibration data exceeded 20 g.

### 3.2. XJTU-SY DataSet

The XJTU-SY dataset is provided by Xi’an Jiaotong University and Changxing Sumyoung Technology Company [29] and the test platform is shown in Figure 4. The test platform consists of a series of components that produce the run-to-failure vibration data under different operating conditions. The radial force generated by the hydraulic loader is applied to the bearing housing of the bearing under test to simulate different working conditions, and the shaft speed is controlled by the motor speed controller. The details of the run-to-failure data for the tested bearings can be found in [29]. The model of the test bearings is LDK UER204, and 32,768 data points in 1.28 s are collected every 1 min with a sampling rate of 25.6 kHz. For this dataset, only horizontal vibration data are used in this study. The accelerated degradation bearing tests are stopped when the vibration amplitude exceeds 20 g for safety, and that moment is considered as the failure time for the bearings.

As shown in Figure 5, the full lifecycle accelerated degradation data of bearing 1 and bearing 2 (PHM 2012 dataset) and bearing 3 and bearing 4 (XJTU-SY dataset) are used to identify the proposed method.

### 3.3. Results and Analysis

#### 3.3.1. Outliers Detection

In order to accurately detect the abnormal segments in each stage, the whole-life vibration signals of the bearings are divided into a number of segments with the same time interval, and the outliers are detected, respectively.

Figure 6a shows the segmented results of the raw vibration signals of bearing 1. At this time, the full lifecycle vibration signals of bearing 1 are divided into segments on average. According to the global detection method of abnormal segments introduced in Section 2, the initial length of the sliding window *w*1 is set to 500 and the initial *k*1 of the local outlier factor is set to 5. The sliding window length and the k are constantly updated in iteration to find the optimum parameters.

After a round of iteration, the global detection results of the abnormal segments are shown in Figure 6b and the signals marked in red are considered as abnormal segments. As can be seen in the local zoom area, the abnormal segments are not only the impulse-type signals (Area 1) that have significant interference with the identification of the degradation stage but also other types of abnormal signal segments (Area 2) with almost no interference with the identification of the degradation stage.

Based on the proposed removing method in Section 2.3, the abnormal segment screening criteria in Formula (11) is adopted to select the impulse-type abnormal segments for the further locating and removing of abnormal impulses.

Figure 7 shows the process of the accurate locating of an abnormal impulse. As can be seen in the local zoom impulse-type abnormal segments in Figure 7a, the detected abnormal segments contain a lot of normal signal data and a very short impulse signal. It is necessary to determine the accurate start and end points of the impulses in order to retain the original normal data to the greatest extent. The local zoom impulse-type abnormal segments in Figure 7a are taken as an example to detect the accurate locating of abnormal impulses.

As show in Figure 7b, the threshold to judge the start point of abnormal impulses is obtained by the kernel density estimation method introduced in Section 2.2. The start point of an abnormal impulse is detected when the sliding local extremum exceeds the threshold for the first time.

Figure 7c shows the detection result of the end point of the abnormal impulse. According to the proposed method in Section 2.2, the gradient threshold is set as 0.3. In this paper, when the gradient g of the envelope fitting fucntion reaches the set threshold, the corresponding position is detected as the end point of the abnormal impulse.

Figure 7d shows the accurate impulse detection results of the abnormal segments. It indicates that the proposed method can accurately identify the location of abnormal impulses and can avoid a large number of normal data being mistakenly removed as abnormal data.

#### 3.3.2. Comparison of Outlier Detection Effects

Figure 8 is the comparison of the outlier detection results. Figure 8a,b,e,f are the outlier detection results based on the local outlier factor [30] of bearing 1, bearing 2, bearing 3 and bearing 4, respectively. A total of 72, 25, 63 and 37 abnormal signal segments are detected in bearing 1, bearing 2, bearing 3 and bearing 4, respectively, containing impulse-type abnormal segments and non-impulse-type abnormal segments. Each impulse-type abnormal segment (marked with red) only contains a lot of normal signal data and a very short impulse signal.

Figure 8c,d,g,h are the outlier detection results based on the proposed method. After the selection of the impulse-type abnormal segments by the proposed removing method in Setion 2.3, the number of the abnormal signal segments is reduced to 54, 23, 47 and 29, in bearing 1, bearing 2, bearing 3 and bearing 4, respectively. Furthermore, adopting the proposed accurate locating method in Setion 2.2, the impulse positions are detected accurately and a large number of normal data avoid being removed.

The important principle of data cleaning is not to significantly change the original data. Compared with the outlier detection method based on the local outlier factor (LOF), the proposed method greatly reduces the removing of normal data. The comparison results of data points after outlier removing are shown in Table 1. It can be seen that the proposed method saves 315,662, 34,052, 9437 and 3834 data points on bearing 1, bearing 2, bearing 3 and bearing 4, respectively.

#### 3.3.3. Comparative of Degradation Stage Recognition

Kurtosis is extremely sensitive to the early failure of the bearing; it is generally considered that mechanical components begin to degrade once the early failure occurs [7]. So, kurtosis is selected as an exemplar indicator to characterize the degradation state and the improved 3σ method is used to identify the degradation point, as shown in Figure 9, Figure 10, Figure 11, Figure 12, Figure 13, Figure 14, Figure 15 and Figure 16. The kurtosis expression is calculated as follows:(14)$$K_{v}=\frac{\frac{1}{n}\sum_{i=1}^{n}{\left(x_{i}-\bar{x}\right)}^{4}}{{\left(\frac{1}{n}\sum_{i=1}^{n}x_{i}^{2}\right)}^{2}}$$
where *x_i_* is the vibration signal data of bearings, *i* = 0, 1, 2, …, *n*; $\bar{x}$ is the mean.

Figure 9, Figure 10, Figure 11 and Figure 12 are the degradation point identification results of bearing 1, bearing 2, bearing 3 and bearing 4 based on raw vibration signals. Figure 13, Figure 14, Figure 15 and Figure 16 are the degradation point recognition results of bearing 1, bearing 2, bearing 3 and bearing 4 after outlier cleaning.

After outlier cleaning, the stability of the health indicators and the robustness of the degraded point identification have been significantly improved.

(1)Stability comparison of health indicator

As can be seen in Figure 9b and Figure 10b, in the normal stage without bearing failure, the degradation health indicator kurtosis is generally kept at a low level and the amplitude fluctuates in a small range. When the bearings begin to degrade, the amplitude of kurtosis will increase significantly. This shows that kurtosis is an effective health indicator of bearing degradation. Figure 11b and Figure 12b show the health indicators of bearing 3 and bearing 4, from which it can be observed that due to the great influence of outliers in the normal stage, the variation range of health indicators becomes smaller when the bearing is really degraded. However, the random impulse-type outliers will cause great interference to the amplitude of kurtosis. Therefore, the impulse-type outliers will reduce the stability of the health index. After outlier cleaning, as shown in Figure 13b, Figure 14b, Figure 15b and Figure 16b, the stability of the kurtosis indicator has been significantly improved. At this time, the amplitude of the health index will not be disturbed by the outliers, which provides important help for health monitoring and degradation stage identification.

(2)Robustness comparison of degraded point identification

According to Formula (12), in the 3*σ* method for degradation stage identification, the mean μ and standard deviation σ of the reference signal are the key parameters to determining the degradation point. The method with good performance should be robust to the selection of the reference signal, that is, for different reference signals, as long as they are contained in the normal stage, the degradation point should be identified at the similar positions.

The robustness of the degradation point identification before and after outlier cleaning with three reference signals is compared in Figure 9, Figure 10, Figure 11, Figure 12, Figure 13, Figure 14, Figure 15 and Figure 16. These three reference signals are the earliest 15%, 20%, 30% data points of the full lifecycle degradation data. The corresponding three degradation points are DP1, DP2 and DP3.

As shown in Figure 9, Figure 10, Figure 11 and Figure 12, before outlier cleaning, when different reference signals are selected, the degradation points are identified in places with large time differences. That means the selection of the reference signal will cause significant interference to the identification of the degradation points, and the robustness of the 3*σ* method without outlier cleaning is very poor.

Conversely, after outliers are cleaned, as shown in Figure 13, Figure 14, Figure 15 and Figure 16, the identified three degradation points (DP1, DP2, DP3) corresponding to the earliest 15%, 20% and 30% data points have small time errors. That means the robustness of the 3*σ* method after outlier cleaning increases significantly. Moreover, after the identified degradation point, the vibration amplitude increases significantly, which verifies the accuracy of the degradation point identification.

Furthermore, to identify the effectiveness of the proposed method, the degradation point identification results of the proposed method are compared with those of the methods proposed by Li et al. [7] and Pan et al. [10]. The health indicators in Li’s approach and our approach are kurtosis characteristics, while the health indicator in Pan’s approach is relative root mean square. These reference signals for the three approaches are the earliest 20% data points of the full lifecycle degradation data. The degradation point identification results of the four bearings are shown in Figure 17, Figure 18, Figure 19 and Figure 20, respectively, and the degradation point identification values are given in Table 2.

Figure 17 plots the corresponding degradation point identification results of bearing 1 by the three approaches. Figure 17a,d are the raw vibration signals and the health indicators of Li’s approach, respectively. Figure 17b,e are the raw vibration signals and the health indicators of Pan’s approach, respectively. Figure 17c,f are the vibration signals and the health indicators after outlier cleaning of our approach, respectively. As can be seen from Figure 17, during the normal stage, there are some abnormal states caused by random noises instead of faults, which interfere with the identification of the degradation points. The degradation points identified by Li’s approach and Pan’s approach have a little time delay. Thanks to the anti-interference ability of our approach, the time when degradation occurs is appropriately selected as the degradation point.

Figure 18, Figure 19 and Figure 20 plot the corresponding degradation point identification results of bearing 2, bearing 3 and bearing 4 by the three approaches; the meanings of the subgraphs of Figure 18, Figure 19 and Figure 20 are shown in Figure 17. For bearing 2, bearing 3 and bearing 4, they behave in abrupt degradation processes. Therefore, it is easier to exactly give the degradation stage of bearing 2, bearing 3 and bearing 4, compared with bearing 1. However, due to the intense interference of outliers during the normal stage, the degradation characteristics of the bearings are obviously weaker than those of the outliers, and it is seen from Figure 18, Figure 19 and Figure 20 that Li’s approach fails to identify the degradation point. In the experiments using Pan, weak changes in the gradient of the health indicators caused misjudgments of the degradation points (the location of the degradation point was determined in advance). Similarly, Pan’s approach also fails to distinguish the normal and the abnormal states. Our approach still performs better than Li’s and Pan’s approaches in identifying the degradation point of bearing 2, bearing 3 and bearing 4.

## 4. Conclusions

In this paper, a degradation stage recognition method of bearings is proposed based on outlier cleaning. The outlier detection method, combining global abnormal segment detection and accurate locating of abnormal impulses, is constructed, realizing the accurate and quick removing of impulse-types outliers that have significant interference with the identification of the degradation points. The main conclusions of this paper are as follows:

The method for locating the start point and the end point of the impulse are constructed to realize the precise identification of the abnormal impulse. While removing abnormal impulses, the normal data are fully retained.

The screening criteria and the iterative removal strategy for abnormal segments are proposed to realize the fast recognition of segments that contain abnormal impulses.

The outlier cleaning avoids the interference of the selection of reference signals when using the 3σ method to identify the degradation stage. The stability of the health indicators and the robustness of the degraded point identification have been significantly improved.

## Figures and Tables

**Figure 1 sensors-22-06480-f001:**
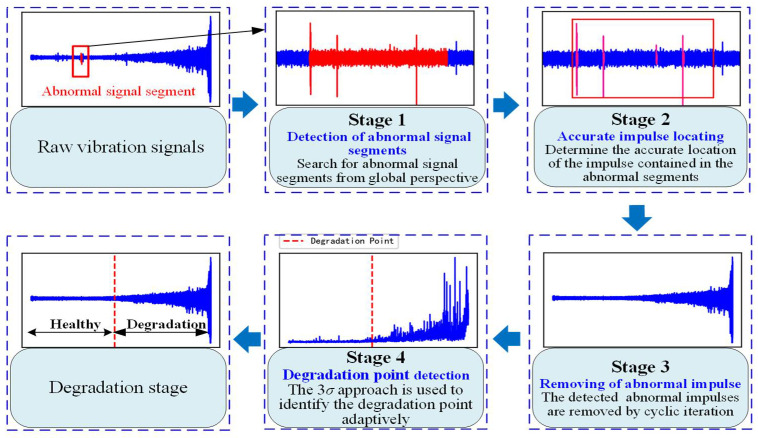
The flow chart of the proposed method.

**Figure 2 sensors-22-06480-f002:**
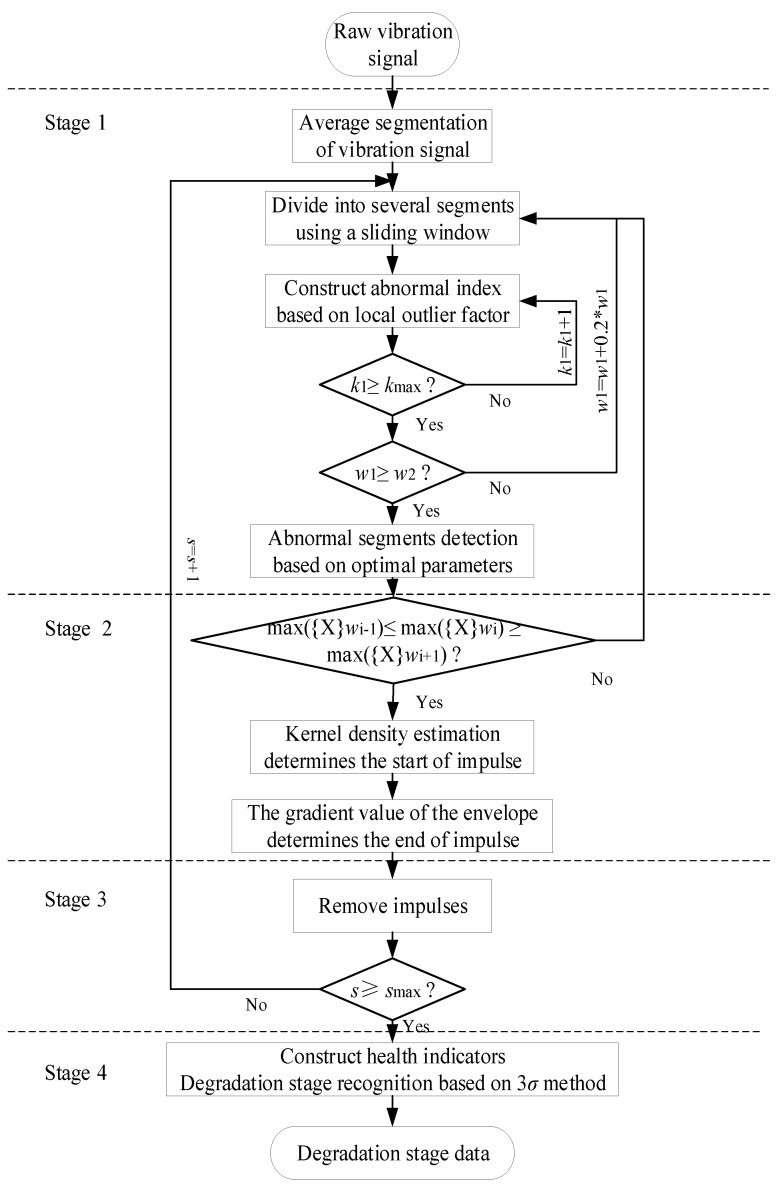
The algorithm flow chart of the proposed method.

**Figure 3 sensors-22-06480-f003:**
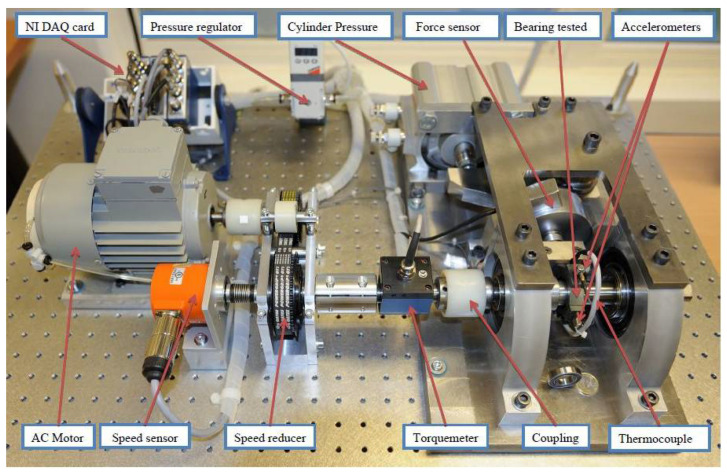
PRONOSTIA test platform.

**Figure 4 sensors-22-06480-f004:**
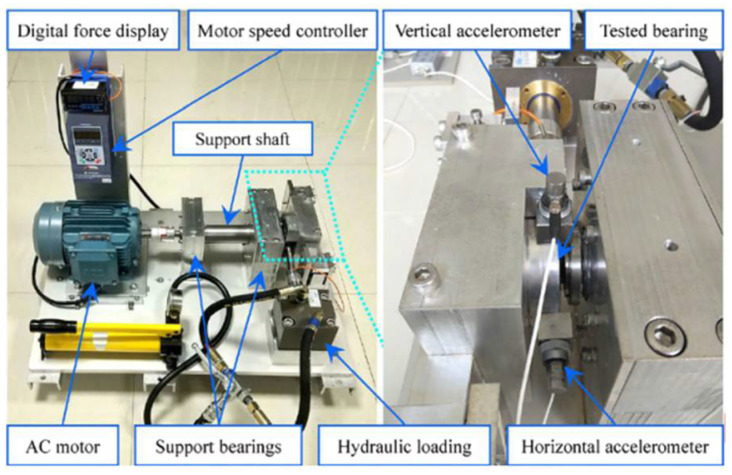
XJTU-SY test platform.

**Figure 5 sensors-22-06480-f005:**
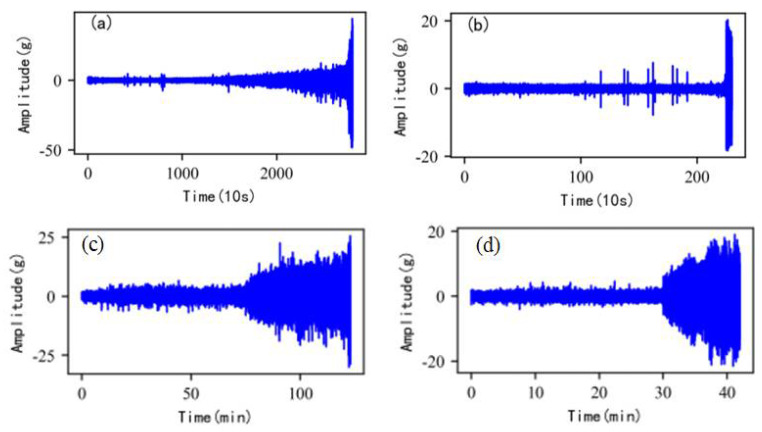
Full lifecycle vibration accelerated degradation data of bearings. (**a**) Bearing 1. (**b**) Bearing 2. (**c**) Bearing 3. (**d**) Bearing 4.

**Figure 6 sensors-22-06480-f006:**
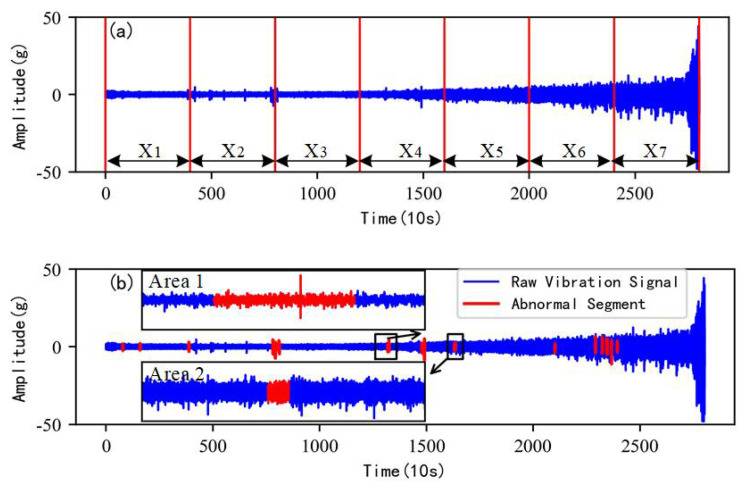
Abnormal signal segments detection of bearing 1. (**a**) Segmented result of the raw vibration signal. (**b**) An iteration of abnormal signal segment detection.

**Figure 7 sensors-22-06480-f007:**
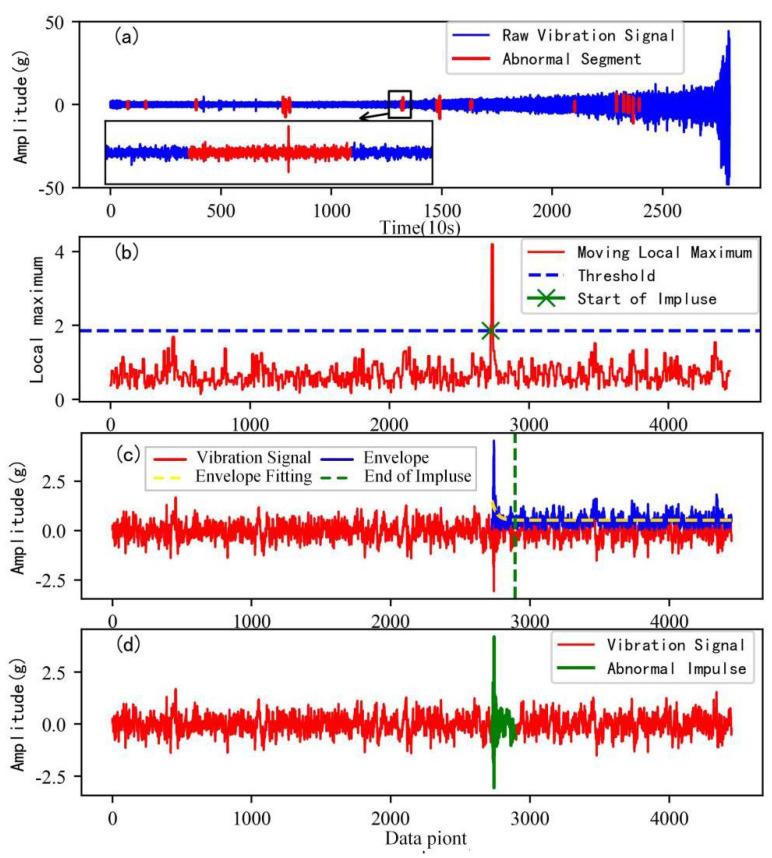
Accurate locating of abnormal impulse of bearing 1. (**a**) Detection of abnormal signal segments; (**b**) the start point of the abnormal impulse; (**c**) the end point of the abnormal impulse; (**d**) accurate locating of abnormal impulse.

**Figure 8 sensors-22-06480-f008:**
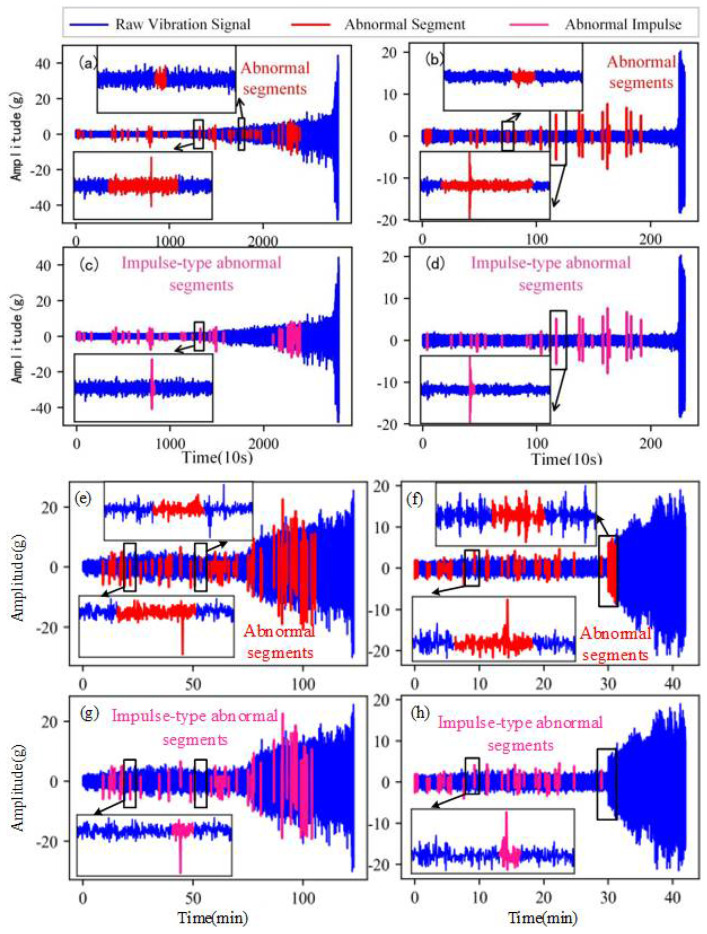
Comparison of outlier detection effects. (**a**) Detection result by proposed method of bearing 1. (**b**) Detection result by proposed method of bearing 2. (**c**) detection result based on local outlier factor of bearing 1. (**d**) Detection result based on local outlier factor of bearing 2. (**e**) Detection result by proposed method of bearing 3. (**f**) Detection result by proposed method of bearing 4. (**g**) Detection result based on local outlier factor of bearing 3. (**h**) Detection result based on local outlier factor of bearing 4.

**Figure 9 sensors-22-06480-f009:**
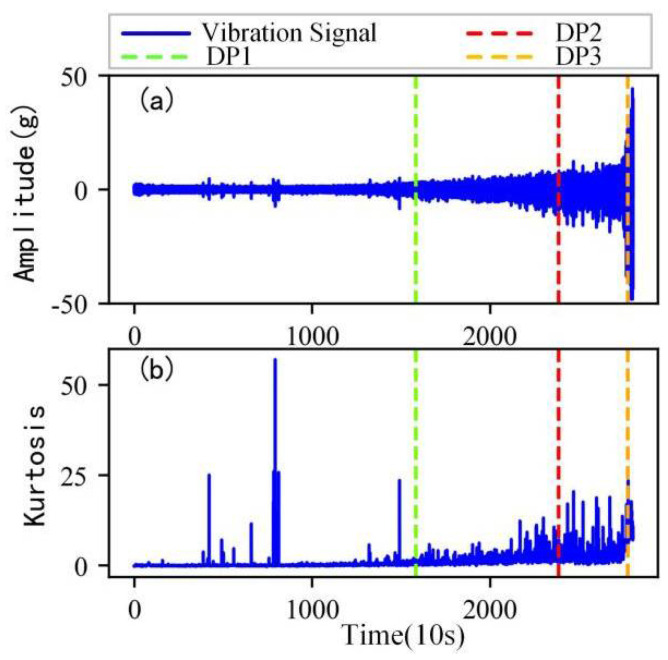
Degradation point identification of bearing 1 before outlier cleaning. (**a**) Vibration signal. (**b**) Kurtosis.

**Figure 10 sensors-22-06480-f010:**
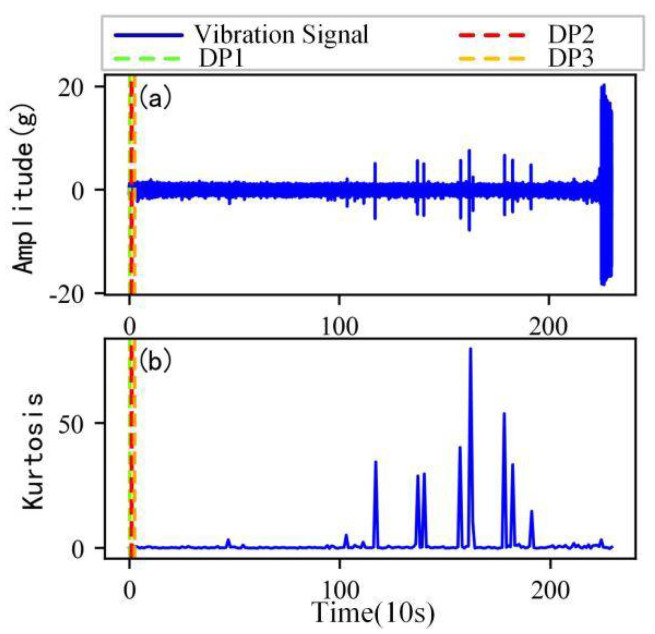
Degradation point identification of bearing 2 before outlier cleaning. (**a**) Vibration signal. (**b**) Kurtosis.

**Figure 11 sensors-22-06480-f011:**
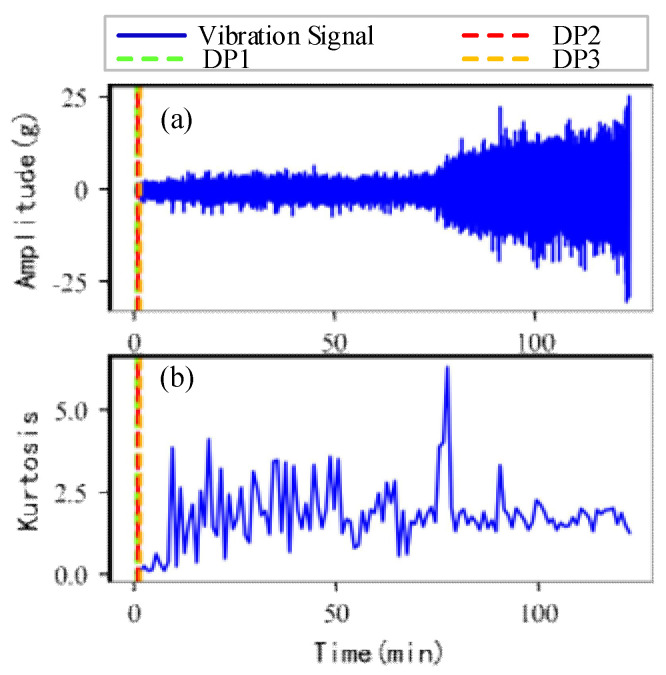
Degradation point identification of bearing 3 before outlier cleaning. (**a**) Vibration signal. (**b**) Kurtosis.

**Figure 12 sensors-22-06480-f012:**
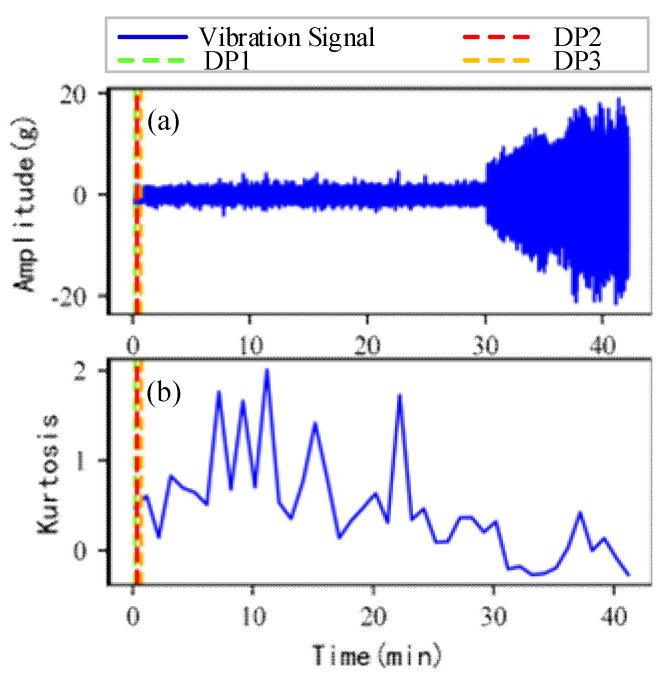
Degradation point identification of bearing 4 before outlier cleaning. (**a**) Vibration signal. (**b**) Kurtosis.

**Figure 13 sensors-22-06480-f013:**
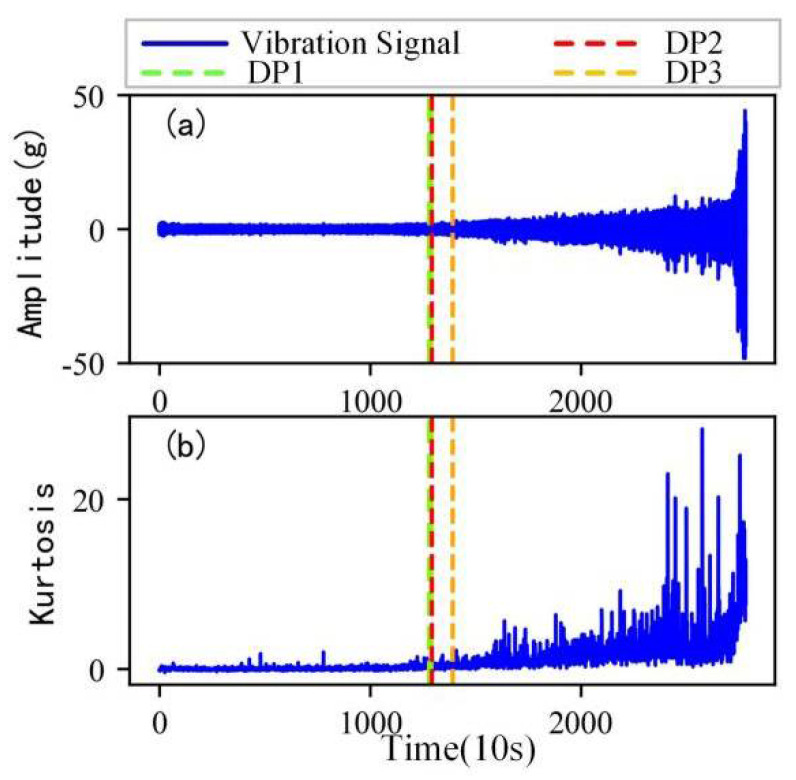
Degradation point identification of bearing 1 after outlier cleaning. (**a**) Vibration signal. (**b**) Kurtosis.

**Figure 14 sensors-22-06480-f014:**
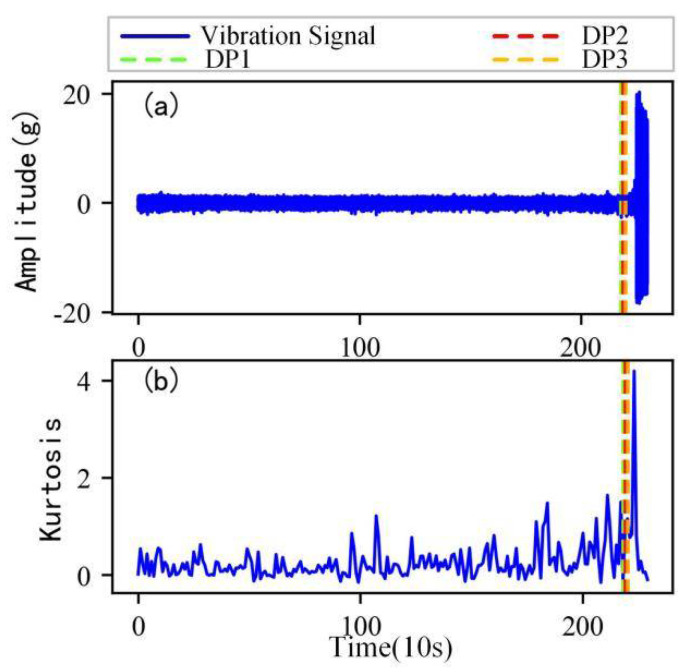
Degradation point identification of bearing 2 after outlier cleaning. (**a**) Vibration signal. (**b**) Kurtosis.

**Figure 15 sensors-22-06480-f015:**
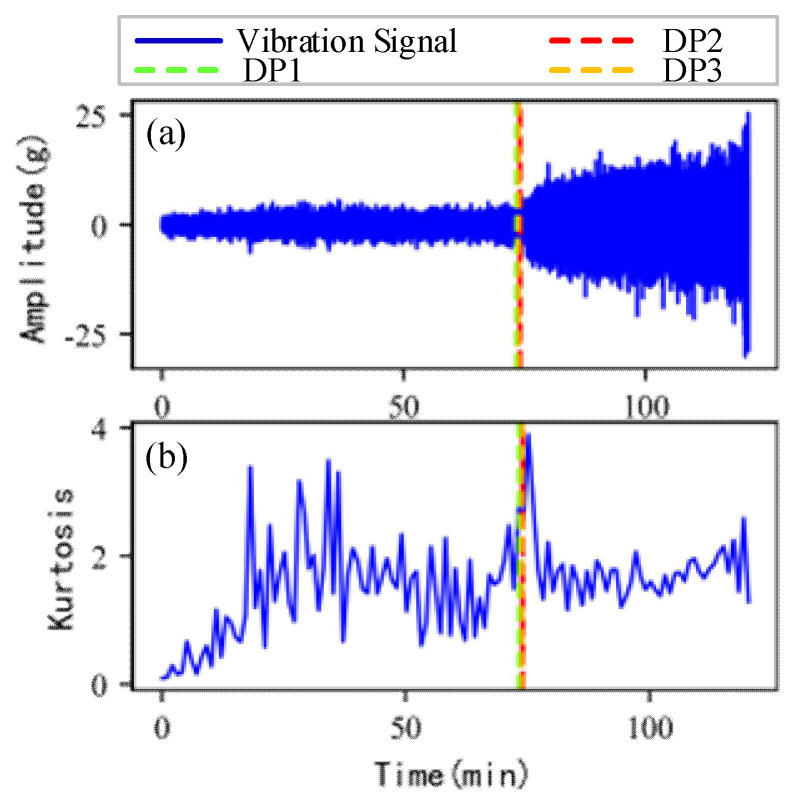
Degradation point identification of bearing 3 after outlier cleaning. (**a**) Vibration signal. (**b**) Kurtosis.

**Figure 16 sensors-22-06480-f016:**
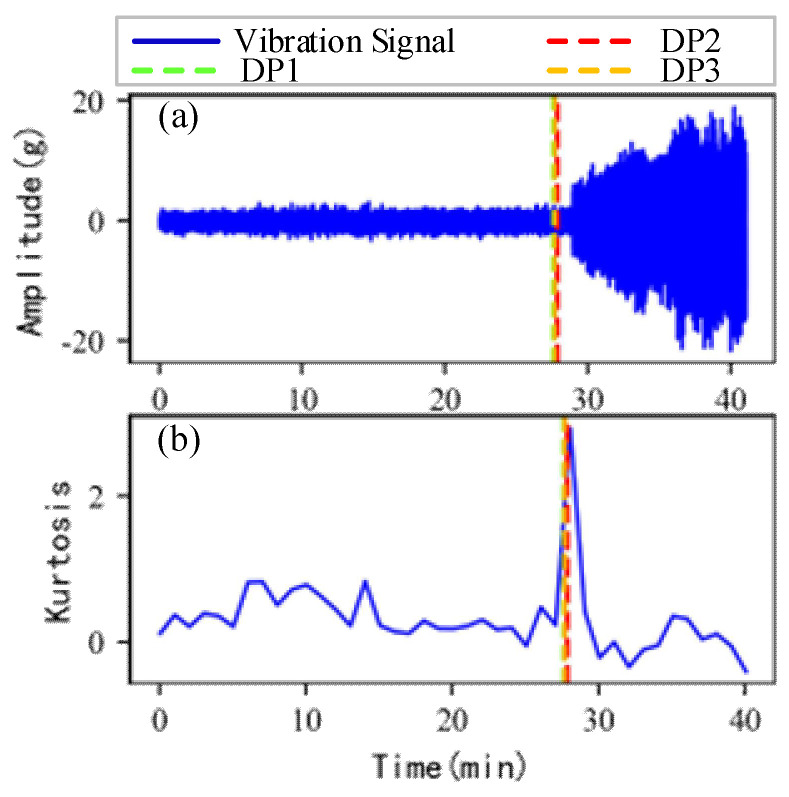
Degradation point identification of bearing 4 after outlier cleaning. (**a**) Vibration signal. (**b**) Kurtosis.

**Figure 17 sensors-22-06480-f017:**
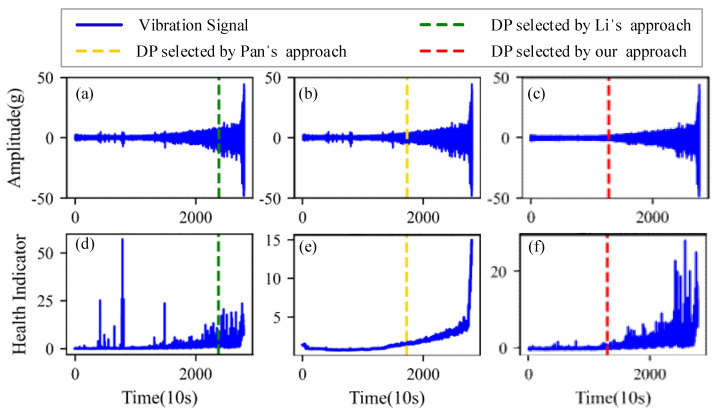
Degradation point identification results of bearing 1. (**a**) Li’s approach. (**b**) Pan’s approach. (**c**) Our approach. (**d**) Li’s approach. (**e**) Pan’s approach. (**f**) Our approach.

**Figure 18 sensors-22-06480-f018:**
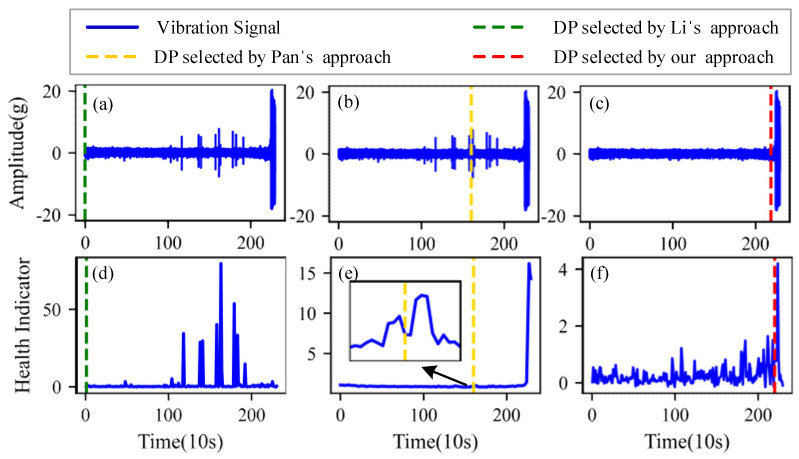
Degradation point identification results of bearing 2. (**a**) Li’s approach. (**b**) Pan’s approach. (**c**) Our approach. (**d**) Li’s approach. (**e**) Pan’s approach. (**f**) Our approach.

**Figure 19 sensors-22-06480-f019:**
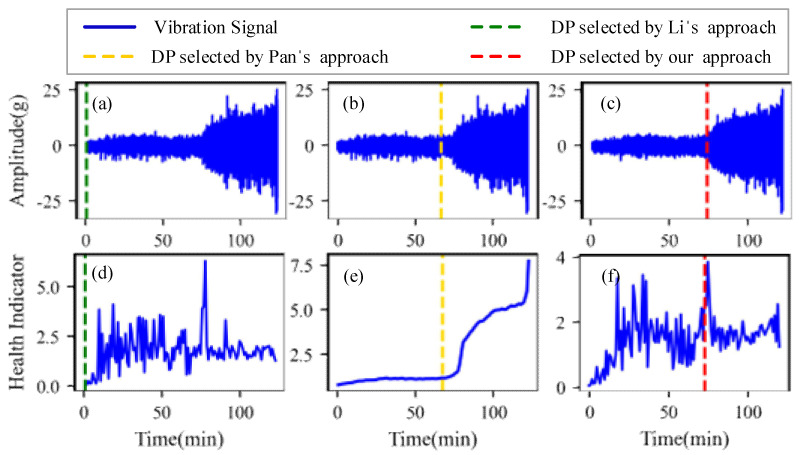
Degradation point identification results of bearing 3. (**a**) Li’s approach. (**b**) Pan’s approach. (**c**) Our approach. (**d**) Li’s approach. (**e**) Pan’s approach. (**f**) Our approach.

**Figure 20 sensors-22-06480-f020:**
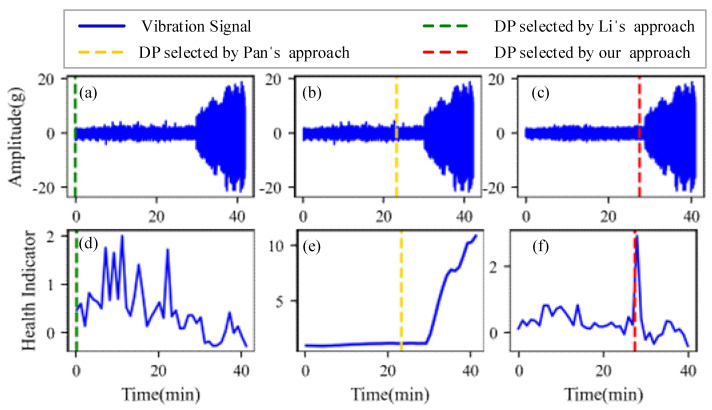
Degradation point identification results of bearing 4. (**a**) Li’s approach. (**b**) Pan’s approach. (**c**) Our approach. (**d**) Li’s approach. (**e**) Pan’s approach. (**f**) Our approach.

**Table 1 sensors-22-06480-t001:** Comparative statistics of vibration data points by outlier processing.

	Data Point			
	Raw Vibration Signal	Method Based on LOF	Proposed Method	Savings
Bearing 1	7,175,680	6,803,081	7,118,743	315,662
Bearing 2	588,800	552,524	586,576	34,052
Bearing 3	314,880	299,469	308,906	9437
Bearing 4	107,520	100,125	103,959	3834

**Table 2 sensors-22-06480-t002:** Comparison results of degradation point identification by different methods of two bearings.

Case	Li’s Approach	Pan’s Approach	Our Approach
Bearing 1	23,810 s	17,230 s	12,910 s
Bearing 2	0 s	1600 s	2190 s
Bearing 3	0 min	67 min	73 min
Bearing 4	0 min	23 min	27 min

## Data Availability

Not applicable.

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
