# Peer review of "An Outlier Cleaning Based Adaptive Recognition Method for Degradation Stage of Bearings"

_sensors, 2022, doi:10.3390/s22176480_

Round 1

Reviewer 1 Report

1. It is recommended that the introduction specifically expatiate the background and research status of the identification of the degradation state of the bearing; In addition, it is recommended that the research status part be specific to the two aspects 1) detection, positioning and removing of abnormal signal, 2) identification methods of bearing degradation status; then concluding the research results and existing problems.

2. It is recommended to create a table to display the data value in section 3.2; including the feature set, local outlier factors of each round of iterative updates, initial threshold and gradient threshold, et al.

3. It is recommended to give an expression of the kurtosis for section 3.2.3.

Author Response

The authors are grateful to receive comments from Reviewer #1. Thank you very much for giving such detailed suggestions for changes. A major rewriting of the paper has been undertaken and responses to the comments are shown as follows.

Reviewer 2 Report

The manuscript entitled “An outlier cleaning based adaptive recognition method for degradation stage of bearings” is relatively well written, with just some typological and grammatical errors, requiring moderate English review. The theme is relevant, the idea of proposing more effective and optimized methods for fault detection in rolling bearings is interesting and the application meets the current needs of improvement in this field.

Some comments regarding the paper:

1)   When comparing the proposed method with the methods in the literature, only 1 case is compared. To make the comparison more robust, it would be interesting to compare it with at least one more case.

2)   Authors should clarify the main novelty of the paper and the real contribution to the area.

Author Response

The authors are grateful to receive comments from Reviewer #2. Thank you very much for giving such detailed suggestions for changes. A major rewriting of the paper has been undertaken and responses to the comments are shown as follows.
